# Single-molecule visualization of fast polymerase turnover in the bacterial replisome

Jacob S Lewis[1,2†], Lisanne M Spenkelink[1,2,3†], Slobodan Jergic[1,2], Elizabeth A Wood[4], Enrico Monachino[1,2,3], Nicholas P Horan[1,2], Karl E Duderstadt[3,5,6], Michael M Cox[4], Andrew Robinson[1,2], Nicholas E Dixon[1,2*], Antoine M van Oijen[1,2*]

[1]Centre for Medical and Molecular Bioscience, University of Wollongong, Wollongong, Australia; [2]Illawarra Health and Medical Research Institute, University of Wollongong, Wollongong, Australia; [3]Zernike Institute for Advanced Materials, University of Groningen, Groningen, The Netherlands; [4]Department of Biochemistry, University of Wisconsin-Madison, Madison, United States; [5]Max Planck Institute of Biochemistry, Martinsried, Germany; [6]Physik Department, Technishche Universität München, Garching, Germany

**Abstract** The *Escherichia coli* DNA replication machinery has been used as a road map to uncover design rules that enable DNA duplication with high efficiency and fidelity. Although the enzymatic activities of the replicative DNA Pol III are well understood, its dynamics within the replisome are not. Here, we test the accepted view that the Pol III holoenzyme remains stably associated within the replisome. We use in vitro single-molecule assays with fluorescently labeled polymerases to demonstrate that the Pol III* complex (holoenzyme lacking the $\beta_2$ sliding clamp), is rapidly exchanged during processive DNA replication. Nevertheless, the replisome is highly resistant to dilution in the absence of Pol III* in solution. We further show similar exchange in live cells containing labeled clamp loader and polymerase. These observations suggest a concentration-dependent exchange mechanism providing a balance between stability and plasticity, facilitating replacement of replisomal components dependent on their availability in the environment.

*For correspondence: nickd@uow.edu.au (NED); vanoijen@uow.edu.au (AMvO)

†These authors contributed equally to this work

## Introduction

The *Escherichia coli* replisome requires participation of 13 different proteins. Ten of them form the DNA polymerase III (Pol III) holoenzyme (HE), which is arranged into three functionally distinct and stably bound subassemblies (*Figure 1A*): αεθ forms the Pol III core that has DNA polymerase activity; $\beta_2$ is the sliding clamp needed for stable association with the primer–template DNA; and $\tau_n\gamma_{(3-n)}\delta\delta'\psi\chi$ (where n = 2 or 3 in the HE) is the clamp loader complex (CLC) that loads $\beta_2$ onto DNA and is the central organizer of the replisome (*Robinson and van Oijen, 2013*; *Duderstadt et al., 2014*; *Lewis et al., 2016*). The CLC interacts with two or three Pol III cores via the α–τ interaction, forming stable complexes termed Pol III* (i.e. HE lacking only the sliding clamp). Pol III* ensures the organization of the cores needed for coordinated DNA synthesis on the two template strands (*Wu et al., 1992*; *Onrust et al., 1995*) and is essential for cell survival (*Blinkova et al., 1993*). Although physical coupling of leading and lagging strand cores in one HE particle requires the lagging strand polymerase to undergo cycles of release and rebinding from one Okazaki fragment to the next, the molecular mechanisms underlying its cycling are still debated (*Dohrmann et al., 2011*). There is, however, consensus that Pol III is

reused rather than replaced for successive Okazaki fragment synthesis (*Tanner et al., 2008*, *2009*, *2011*; *Yao et al., 2009*; *Georgescu et al., 2011*). Thus, the replisome is believed to be a highly stable entity.

The key observations that support efficient Pol III recycling derive from in vitro replication assays in the absence of free polymerase (*Yao et al., 2009*; *Tanner et al., 2011*; *Georgescu et al., 2011*) and are consistent with the high stability of the α–τ interaction that binds cores to the CLC ($K_D$ = 0.3 nM; $t_{1/2}$ = 29 min in 300 mM NaCl) (*Jergic et al., 2007*). Nevertheless, the introduction of high concentrations of catalytically dead Pol III* (still able to bind primed DNA) inhibits ongoing replication (*Yuan et al., 2016*). Reconciling these different observations, we here demonstrate the presence of a novel exchange mechanism that allows Pol III* to remain stably associated with the replisome under conditions of high dilution, yet facilitates rapid exchange at nanomolar concentrations.

## Results

### In vitro single-molecule observation of Pol III dynamics

We use a single-molecule approach to directly visualize the dynamics of Pol III complexes at the replication fork (*Mok and Marians, 1987*; *Tanner et al., 2009*). A rolling-circle DNA amplification scheme is used to observe highly processive DNA synthesis in real time, while imaging Pol III complexes entering and leaving the replisome. Using the minimal set of 12 proteins required to support coupled leading and lagging strand synthesis, we allow active replisomes to self assemble onto preformed replication forks (*Tanner et al., 2008*, *2009*). A 5′-flap within a 7.2 kb double-stranded (ds) circular DNA substrate is anchored to the surface of a microfluidic flow cell and replication is initiated by introducing a laminar flow of buffer with the components required for coupled leading and lagging strand synthesis (*Figure 1B*). As replication proceeds, the newly synthesized leading strand becomes part of the circle and later acts as a template for lagging strand synthesis. With the lagging strand attached to the surface and the continuously growing DNA product stretched in the buffer flow, the dsDNA circle moves away from the anchor point. Replication is visualized by real-time near-TIRF fluorescence imaging of stained dsDNA (*Figure 2A*, *Figure 2—figure supplement 1*). This strategy allows quantification of the rates of individual replisomes and their processivities (*Figure 2B*).

We fluorescently labeled the Pol III α subunit following its fusion to a SNAP tag (*Figure 2—figure supplement 2*) and covalently coupled it separately in >80% yields to red and green fluorophores. Fluorescently labeled Pol III cores were reconstituted from individual SNAP-α, ε and θ subunits and isolated chromatographically (*Tanner et al., 2008*), then assembled into single-color Pol III*s in situ with separately isolated τ₃-CLC (*Tanner et al., 2008*). The labeled Pol III*s were active in coupled DNA replication, producing Okazaki fragments of similar sizes to wild-type polymerase (*Figure 2—figure supplement 3*).

A kymograph (*Figure 2C*) shows the fluorescence of the red Pol III* during rolling-circle replication; it supports replication at rates similar to the untagged wild-type enzyme (*Figure 2D*). Simultaneous imaging of the stained DNA and red Pol III* shows that the polymerase spot is located at the tip of the growing DNA, confirming that the labeled Pol III is a functional component of reconstituted replisomes (*Figure 2—figure supplement 4*). We also observe Pol III that remains bound to the DNA on the lagging strand behind the replication fork, evident as horizontal lines in *Figure 2C*. We reasoned that these correspond to polymerases bound to the 3′-termini of Okazaki fragments. We repeated the experiment in the presence of Pol I and/or DNA ligase; Pol I replaces RNA primers with DNA and ligase seals the remaining nick. In the presence of Pol I (with or without ligase), Pol III binding behind the replisome is no longer observed (*Figure 2E*), consistent with Pol I efficiently displacing Pol III during Okazaki fragment maturation.

Surprisingly, the fluorescent Pol III at the growing tip of the rolling circle is highly resistant to photobleaching. Its fluorescence in the replisome has a much longer lifetime compared to that of labeled Pol III cores immobilized on a surface and subjected to the same excitation intensity (*Figure 2F*). Since the experiments in *Figure 2* are performed with 6.7 nM Pol III* in solution, this observation suggests that the polymerase exchanges into the replisome from solution to replace photobleached Pol III.

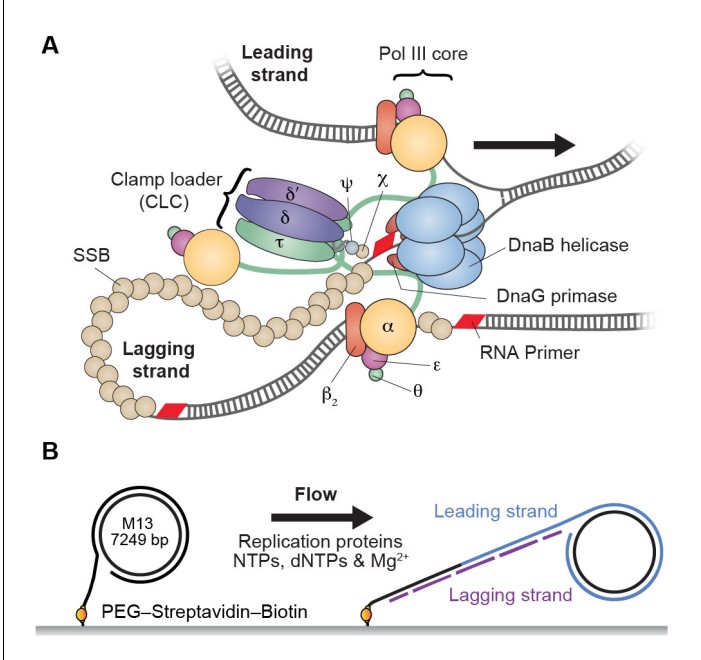

**Figure 1.** Single-molecule rolling-circle replication assay. (**A**) Canonical view of the organization of the *E. coli* replication fork. The DnaB helicase encircles the lagging strand, facilitates unwinding of dsDNA through ATP hydrolysis, and recruits DnaG primase for synthesis of RNA primers that initiate synthesis of 1–2 kb Okazaki fragments on the lagging strand. The extruded single-stranded (ss) DNA is protected by ssDNA-binding protein, SSB. The Pol III holoenzyme (HE) uses the ssDNA of both strands as a template for coupled, simultaneous synthesis of a pair of new DNA duplex molecules. The $\beta_2$ sliding clamp confers high processivity on the Pol III HE by tethering the αεθ Pol III core onto the DNA. The clamp loader complex (CLC) assembles the $\beta_2$ clamp onto RNA primer junctions. Up to three Pol III cores interact with the CLC through its τ subunits to form the Pol III* complex, and the τ subunits also interact with DnaB, thus coupling the Pol III HE to the helicase. (**B**) Schematic representation of the experimental design. 5'-Biotinylated M13 DNA is coupled to the passivated surface of a microfluidic flow cell through a streptavidin linkage. Addition of the *E. coli* replication proteins and nucleotides initiates DNA synthesis. The DNA products are elongated hydrodynamically by flow, labeled with intercalating DNA stain, and visualized using fluorescence microscopy (*Figure 1—figure supplement 1*).

The following figure supplement is available for figure 1:

**Figure supplement 1.** Schematic overview of the single-molecule fluorescence microscope.

## Exchange of Pol III* complexes in vitro

To characterize the dynamic behavior of Pol III at the fork and directly visualize its exchange in real time, we used mixtures of red and green Pol III*s. To demonstrate that green Pol III cores in a Pol III* complex do not exchange with the red ones from another Pol III*, we combined them in a 1:1 ratio for 1 hr at 37°C (*Figure 3A*), then imaged the mixture on the surface of a coverslip at the single-molecule level (*Figure 3B*, *Figure 3—figure supplement 1*). Consistent with the stable interaction between α in the core and τ in the CLC, exchange of Pol III cores was not observed. It remained possible, however, that the nature and strength of the α–τ interaction is different at a step in lagging strand replication that involves exchange of Pol III cores within the Pol III* complex (*Leu et al., 2003*). To test this possibility, we mixed pre-assembled red and green Pol III* complexes in a 1:1 ratio and used them in a bulk rolling-circle replication experiment in an eightfold molar excess over 5'-biotinylated flap-primed dsDNA template. Under these conditions, most Pol III*s will have participated in replication at the fork, as long leading strand and shorter lagging strand products are generated (*Figure 3—figure supplement 2*). Next, the newly synthesized DNA was removed from proteins by its immobilization on streptavidin beads, and subsequent single-molecule imaging of the released protein fraction on the surface of a coverslip showed no co-localization of red and green

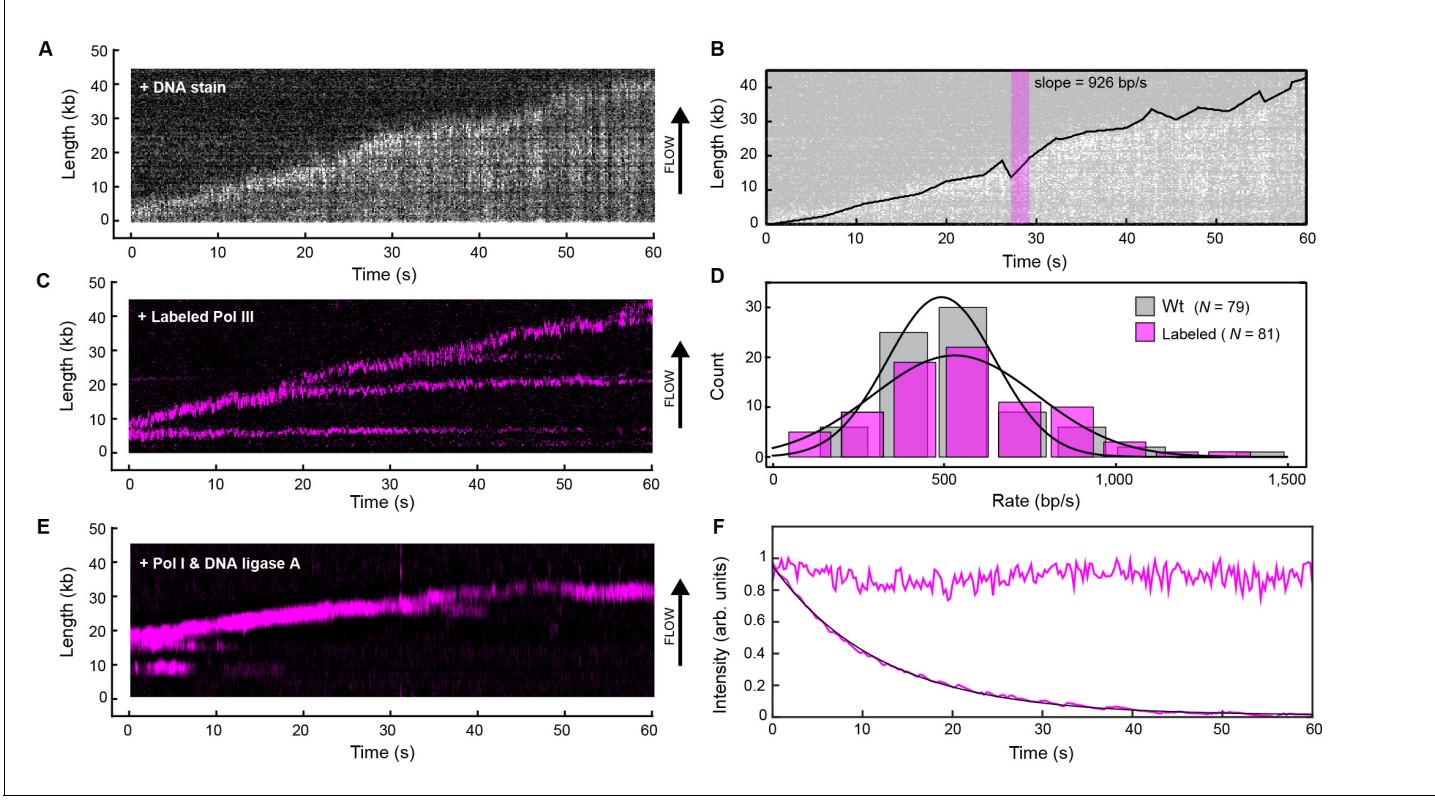

**Figure 2.** Real-time fluorescence imaging of coupled DNA replication. (**A**) Kymograph of an individual DNA molecule undergoing coupled leading and lagging strand replication. The gray scale indicates the fluorescence intensity of stained DNA. (**B**) Single-molecule trajectory obtained from the kymograph in (**A**), used to quantify the rates and processivities of replication events. The magenta box represents an example line segment used to determine rates. (**C**) Kymograph of the dynamics of red-labeled Pol IIIs on an individual DNA molecule. The Pol III moves with the replisome in the direction of flow as it elongates the DNA, visible as a bright magenta spot moving away from the surface anchor point. Additional Pol IIIs are left behind the moving replisome, seen as horizontal lines on the kymograph. (**D**) Histograms of the rate of replication for wild-type Pol III (492 ± 23 bp/s) and red Pol III (561 ± 27 bp/s) fit to Gaussian distributions. (**E**) Kymograph of the distribution of red Pol III on an individual DNA molecule in the presence of 150 nM Pol I and 100 nM DNA ligase. Prolonged Pol III spots behind the replisome are no longer observed due to the action of Pol I in Okazaki fragment processing. (**F**) Fluorescence intensity as a function of time of individual red Pol IIIs immobilized on the surface of a coverslip (lower trace; black line is an exponential fit with lifetime = 14.1 ± 0.4 s), and of the replisomal spot in (**C**) (upper trace). The fluorescence lifetime of red Pol III at the replisome is much longer than the photobleaching lifetime of the dye. The errors represent the standard errors of the mean.

The following figure supplements are available for figure 2:

**Figure supplement 1.** Representative field of view of SYTOX orange-stained dsDNA from the single-molecule rolling-circle DNA replication assay.

**Figure supplement 2.** Separation of proteolytic fragments of SNAP-α from full-length SNAP-α.

**Figure supplement 3.** Comparison of activities of wild-type and SNAP-labeled Pol III cores.

**Figure supplement 4.** Representative kymograph of simultaneous staining of double-stranded DNA and fluorescence imaging of Pol III in real time.

Pol III cores (*Figure 3C*). This result confirms that the functional unit exchanging at the replication fork is the entire Pol III* complex; the interaction between the τ subunit of the CLC and α of Pol III must remain intact during DNA replication, thus challenging the previously suggested model of a τ processivity switch on the lagging strand (*Leu et al., 2003*).

We visualized exchange of Pol III* at the replication fork by measuring the fluorescence intensity at the replisome spot as a function of time using 1:1 mixtures of red and green Pol III*s (*Figure 4A, B, Figure 4—figure supplement 1*). At a total Pol III* concentration of 6.7 nM, the replisomal spot exhibits fast dynamics displaying both colors (*Video 1*), while at a lower concentration of 0.3 nM,

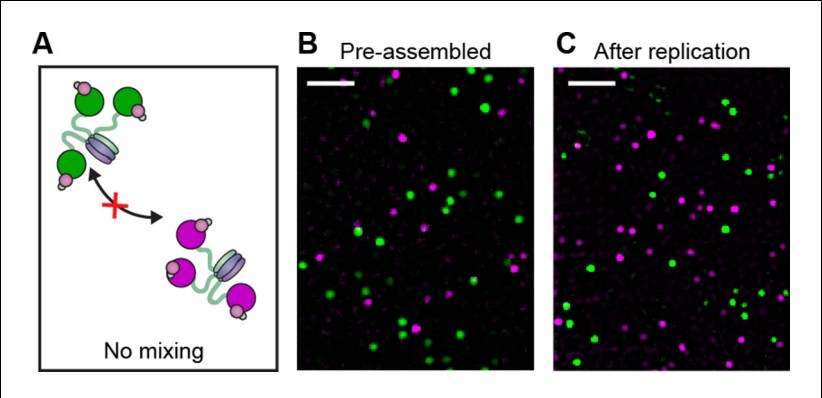

**Figure 3.** Pre-assembled Pol III* complexes do not exchange Pol III core. (**A**) Red and green Pol III* are separately pre-assembled by treatment at 37°C for 15 min (30 nM Pol III core and 10 nM $\tau_3$-CLC). These are then mixed in equal ratios and kept at 37°C for 1 hr prior to dilution to 6 pM Pol III* for imaging. (**B**) Red Pol III* complexes and green Pol III* complexes do not co-localize to produce any white spots as seen in *Figure 3—figure supplement 1*, demonstrating the α–τ interaction within the Pol III* complex remains intact for the duration of the DNA replication assays. (**C**) Pre-assembled red and green Pol III* complexes that have participated in DNA replication (at 3.3 nM of each) do not co-localize, demonstrating that the Pol III cores within a Pol III* do not exchange with cores from other Pol III*s at the replication fork during active DNA synthesis. White scale bars represent 5 μm.

The following figure supplements are available for figure 3:

**Figure supplement 1.** Pol III* complexes of mixed Pol III core composition (1:1) show co-localization.

**Figure supplement 2.** Alkaline gel showing leading- and lagging-strand products using pre-assembled red and green Pol III*s.

---

the dynamics appear slower and distinct exchange events are visible (*Video 2*). The longer persistence of a single color at the lower concentration demonstrates that Pol III* exchange is concentration dependent. Given that Pol III* remains intact on time scales much longer than the duration of our experiment, these observations can only be explained by wholesale exchange of Pol III* at the replication fork.

Our demonstration of rapid exchange of entire Pol III*s, however, seems difficult to reconcile with observations that both leading and lagging strand Pol III cores remain stably associated during coupled DNA replication (*Yao et al., 2009*; *Tanner et al., 2011*; *Georgescu et al., 2011*). Those studies used assays in which replisomes were assembled, replication initiated, and the reactions rapidly diluted to measure the stability of synthesizing replisomes on DNA. To place our observations of dynamic exchange of Pol III* in context of the previous work, we carried out single-molecule pre-assembly replication assays (*Yao et al., 2009*; *Tanner et al., 2011*) using the red Pol III*. In this experiment, the replisome is pre-assembled onto the rolling-circle template in solution. Subsequently, the template is attached to the surface of a flow cell, which is then washed to remove all unbound proteins. Replication is initiated by introduction of a replication solution that omits Pol III* and helicase. Since the absence of free Pol III* in solution makes polymerase exchange impossible, we hypothesized that Pol III would be recycled within the replisome, enabling its sustained participation in processive DNA replication. These conditions indeed support highly processive DNA replication (*Figure 4C*), with synthesis rates and processivities identical to a situation with Pol III* in solution and consistent with values reported previously (*Mok and Marians, 1987*; *McInerney et al., 2007*; *Tanner et al., 2009*; *Yao et al., 2009*) (*Figure 4D,E*). Further, we observed photobleaching without recovery, consistent with the original, pre-assembled Pol III* remaining stably associated within the replisome. As further confirmation of the robustness of the pre-assembled replisome in the absence of competing polymerases and the easy displacement of Pol III* upon challenge, we initiate replication by pre-assembly of replisomes, normally supporting highly processive synthesis, and challenge them with Pol III core. The observation of a sharp reduction in processivity is consistent with the

displacement of the Pol III* from the replication fork by the Pol III cores, which are unable to support coordinated leading and lagging strand synthesis (*Wu et al., 1992*) (*Figure 4—figure supplement 2*).

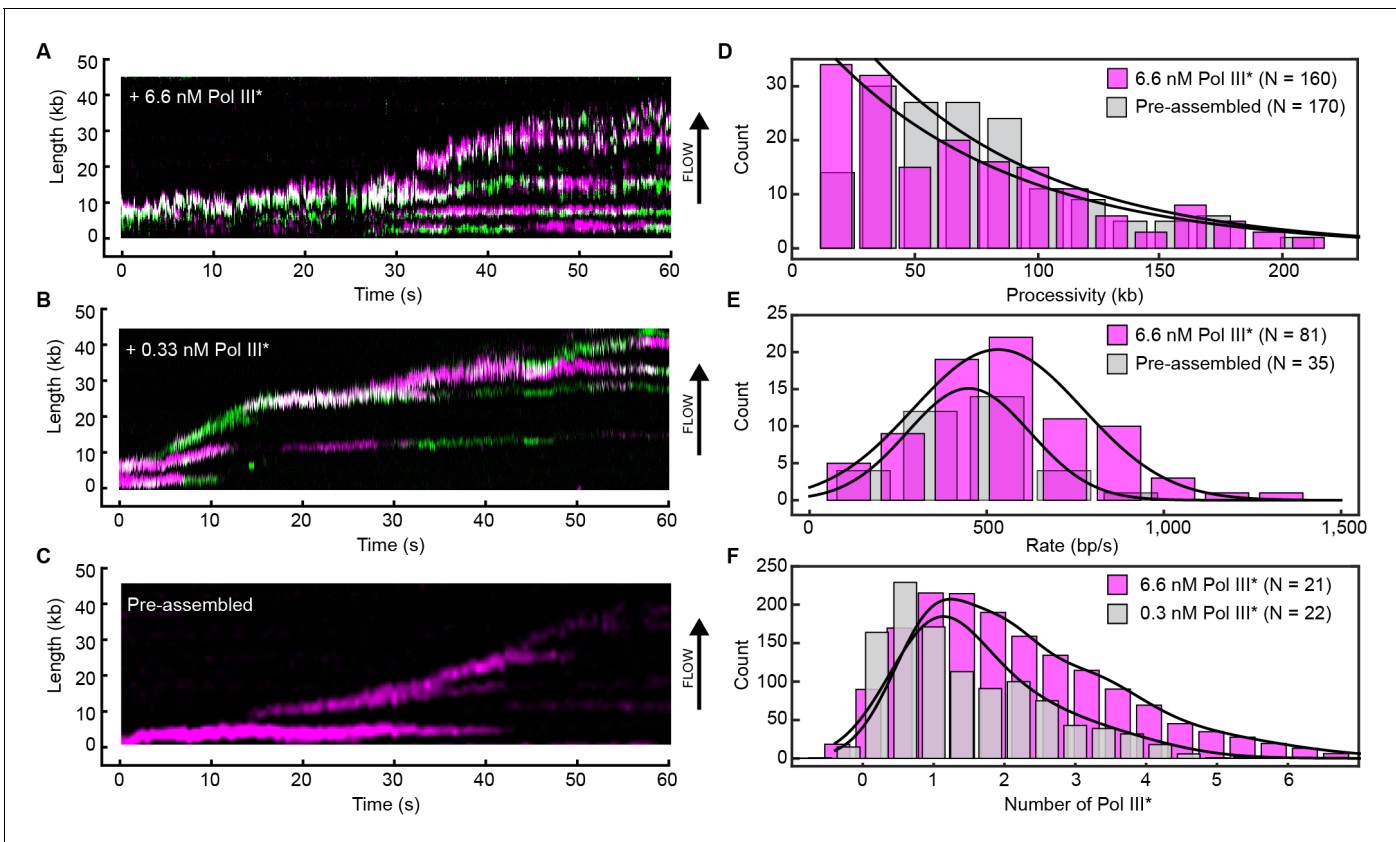

**Figure 4.** Rapid and frequent exchange of Pol III* is concentration dependent. (**A**) and (**B**), Kymographs of the distributions of red Pol III* (magenta) and green Pol III* (green) on an individual DNA molecule at a total Pol III* concentration of 6.7 (**A**) or 0.3 nM (**B**). Co-localization of the two signals is shown as a bright white fluorescent spot. (**C**) Kymograph of a pre-assembled replisome containing red Pol III*. The intensity of the signal from the replisomal spot decreases after a Pol III* is left behind. It subsequently bleaches and the signal does not recover. (**D**) Histograms of the processivity of replication with Pol III* present in solution (73 ± 25 kb) and under pre-assembly conditions (76 ± 26 kb), each fit with a single exponential decay function. The errors represent the error of the fit. (**E**) Histograms of the rates of replication with Pol III* present in solution (561 ± 27 bp/s) and under pre-assembly conditions (445 ± 33 bp/s), each fit to a Gaussian distribution. (**F**) Histograms of the stoichiometry of Pol III* at the replication fork, fit to four (6.7 nM) or three (0.3 nM) Gaussians centered at integral numbers of Pol III* calculated from single Pol III core intensities (see *Figure 4—figure supplement 4*). The black lines represent the sums of these distributions. The errors in (**E**) and (**F**) represent the standard errors of the mean.

The following figure supplements are available for figure 4:

**Figure supplement 1.** Example kymographs.

**Figure supplement 2.** Processivity of replication challenged by Pol III core.

**Figure supplement 3.** Histograms of the stoichiometry of Pol III* at the replication fork.

**Figure supplement 4.** Histograms of the intensity distribution of single Pol III cores.

**Figure supplement 5.** Fluorescence intensity of replicating Pol III* complexes does not change at longer DNA lengths under near-TIRF imaging conditions.

## Quantification of exchange time of Pol III* in vitro

To quantify the concentration-dependent exchange times of Pol III* during coupled DNA replication, we performed in vitro single-molecule FRAP (fluorescence recovery after photobleaching) experiments. We visualized red Pol III* entering and leaving the replication fork at different concentrations using the same rolling-circle amplification scheme as described in *Figure 1B*. Instead of continuous imaging at constant laser power, we periodically bleached all Pol III* at the replication fork using a high-laser power (*Figure 5A*). By bleaching the fluorescence signal of Pol III* complexes, we can monitor the recovery of the florescence signal as unbleached Pol III*s from solution exchange into the replisome (*Figure 5B*). We monitored the recovery of the fluorescence signal and calculated the average intensity after each FRAP pulse over time (*Figure 5C*, *Figure 5—figure supplement 1*). By measuring the single-molecule FRAP of Pol III* over a concentration series spanning four orders of magnitude and fitting the rate of signal recovery, we obtained the characteristic exchange time of Pol III* into active replisomes (*Figure 5D*). At a total Pol III* concentration of 13 nM, the fluorescence signal recovers rapidly (characteristic exchange time, $\tau$ = 1.85 s), while at 30 pM the fluorescence signal is 20-fold slower to recover ($\tau$ = 42 s). These observations are in agreement with our previous two-color experiments, indicating that the rate of exchange is dependent on Pol III* concentration.

## Exchange of Pol III* complexes in live cells

Inspired by our observations of rapid exchange of Pol III* in vitro, we used in vivo single-molecule measurements to determine whether Pol III* exchange also occurs in live *E. coli* cells. We imaged cells in which the clamp loader and core complexes were labeled at their C-termini with yellow and red fluorescent proteins, respectively ($\tau$-YPet, $\varepsilon$-mKate2). As observed previously, these fusions remain fully functional (*Reyes-Lamothe et al., 2010*) (*Figure 6—figure supplement 1*). These cells were immobilized on a (3-aminopropyl)triethoxysilane-treated coverslip and $\tau$-YPet and $\varepsilon$-mKate2 foci were imaged simultaneously. Fluorescent proteins bound to large structures such as the nucleoid diffuse slowly and thus present in our images as diffraction-limited foci, whereas the signal from proteins freely diffusing through the cytosol blurs over the entire cell (*Robinson et al., 2015*). To monitor exchange of polymerase molecules at replisomes, the fluorescence intensity of individual replication foci was tracked over time. We noticed that once the population of fluorescent molecules had become partially nonfluorescent by irreversible photobleaching, instances of synchronized intensity fluctuations of $\tau$-YPet and $\varepsilon$-mKate2 within the replisome foci could be observed (*Figure 6A*). To determine whether these intensity fluctuations were truly correlated, that is, whether they could be explained by the exchange of Pol III*, we used cross-correlation analysis, a powerful unbiased method that enables the calculation of (i) the extent of similarity between two fluctuating signals and (ii) on which timescales that similarity occurs. The average cross-correlation function calculated for 1210 foci in 480 cells showed a clear positive cross-correlation peak, consistent with synchronous exchange of $\tau$-YPet and $\varepsilon$-mKate2 (*Figure 6B*, black line). To show that this peak arises due to protein dynamics, we fixed cells with formaldehyde to arrest all cellular processes and demonstrate the absence of a cross-correlation peak (*Figure 6B*, gray line). To eliminate the possibility of correlated intensity changes due to laser fluctuations, we calculated the cross-correlation function for random pairs of $\tau$-YPet and $\varepsilon$-mKate2 foci within the same field of view, but from different cells. Also here, no cross-correlation peak was detected (*Figure 6B*, red line). Our experimental data cannot be explained by exchange of core only, the uncoupled exchange of both $\varepsilon$ and $\tau$, or even the complete absence of exchange (*Figure 6—figure supplement 2*). In support of Pol III* exchange, a positive

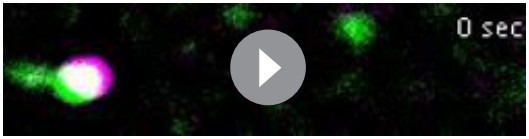

**Video 1.** Movie of red Pol III* (magenta) and green Pol III* (green) on an individual DNA molecule at a total Pol III* concentration of 6.7 nM. This movie was used to generate the kymograph in *Figure 4A*.

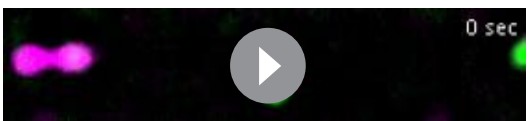

**Video 2.** Movie of red Pol III* (magenta) and green Pol III* (green) on an individual DNA molecule at a total Pol III* concentration of 0.3 nM. This movie was used to generate the kymograph in *Figure 4B*.

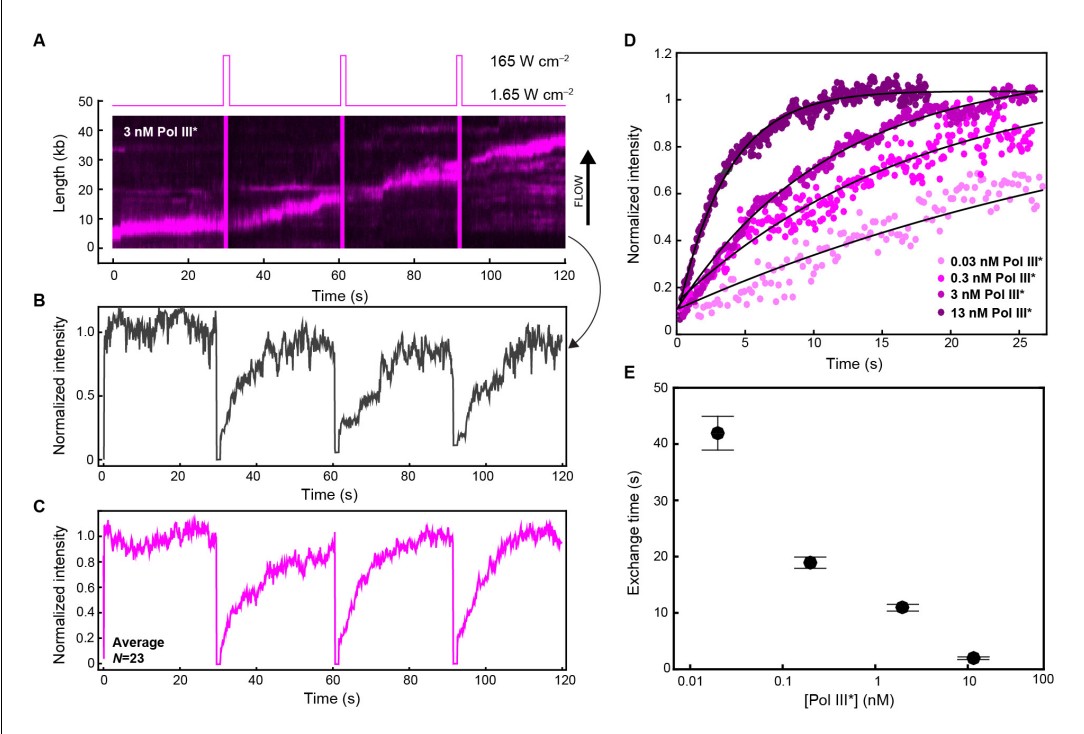

**Figure 5.** Quantification of Pol III* exchange time using single-molecule FRAP. (**A**) (Top panel) Imaging sequence used during the FRAP experiments. Periodically, a FRAP pulse of high-laser power was used to rapidly photobleach all the Pol III* in the field of view. (Bottom panel) A representative kymograph of red Pol III*s at the replication fork. After each FRAP pulse (indicated by the magenta line), all Pol III*s have bleached, but the fluorescence intensity recovers as unbleached Pol III*s exchange into the replisome. (**B**) Normalized intensity over time for an individual replisome in the presence of 3 nM Pol III* in solution. (**C**) The average intensity over time from 23 replisomes with 3 nM Pol III* in solution. (**D**) The three recovery phases in (**C**) were averaged again to give the final averaged normalized intensity over time after a FRAP pulse. This curve was then fit to provide a characteristic exchange time. This was done for four concentrations of Pol III* ranging from 13 to 0.03 nM. (**E**) Exchange time as a function of Pol III* concentration.

The following figure supplement is available for figure 5:

**Figure supplement 1.** Example kymographs for the single-molecule FRAP experiments.

cross-correlation peak can, however, be explained by simultaneous exchange of ε and τ (*Figure 6—figure supplement 2*).

We then calculated the in vivo exchange time by fitting the cross-correlation function with an exponential decay. From this, we found an exchange time of 4 ± 2 s (*Figure 6C*), consistent with measurements performed under similar experimental conditions (*Beattie et al., 2017*). Furthermore, the concentration of τ and ε was determined in the cell (*Figure 6—figure supplement 3*). Similarly to previous observations (*Reyes-Lamothe et al., 2010*), we found under our experimental conditions a total concentration of 72 ± 3 nM of ε and 67 ± 5 nM of τ. Assuming all ε and τ form functional Pol III* complexes within the cell, these concentrations of ε and τ would correspond to ~23 nM Pol III* per cell. These in vivo measurements are consistent with exchange times measured for the highest concentration of Pol III* in vitro (a few seconds at 13 nM). Given these observations, we conclude that Pol III* exchange occurs during coupled DNA replication in vivo.

## Discussion

We conclude that the *E. coli* replisome strikes a balance between stability and plasticity. In the absence of Pol III* in solution, it retains its original polymerase and forms a highly stable complex resistant to dilution. In its presence, Pol III* readily exchanges into the replisome at a rate that is dependent on its concentration. Such a concentration-dependent exchange mechanism seems

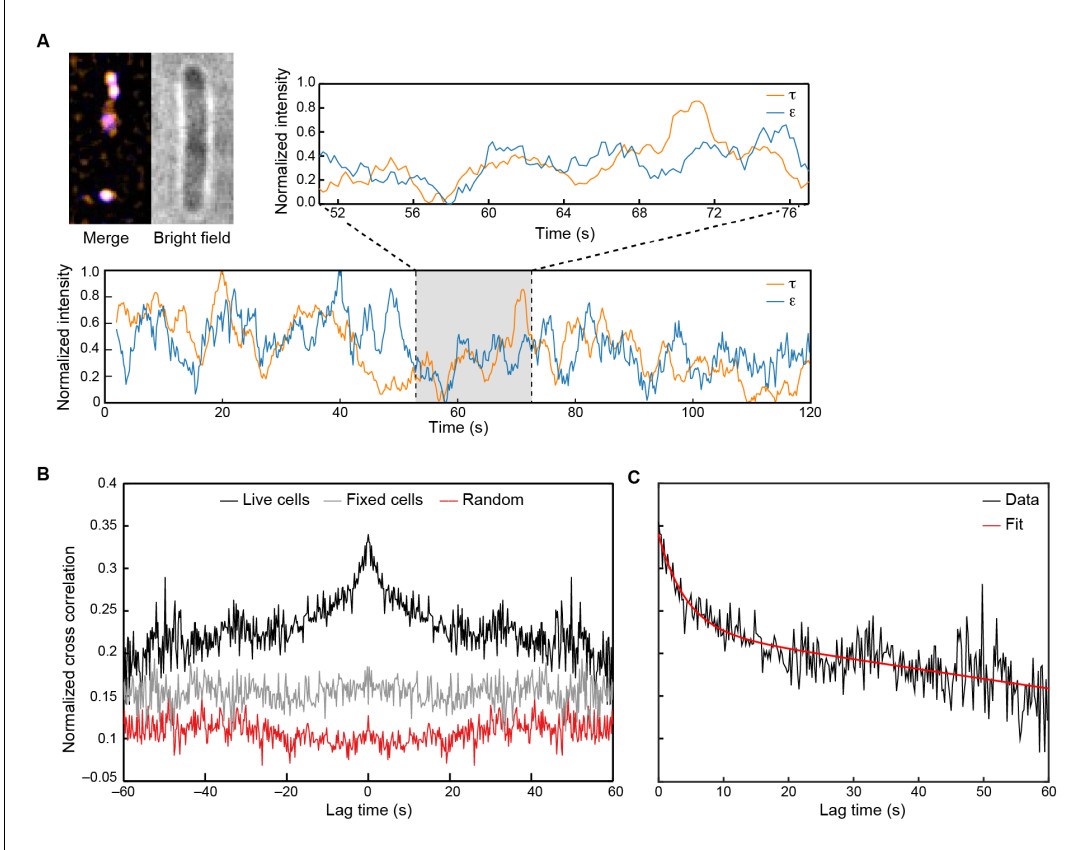

**Figure 6.** Visualization of Pol III* exchange in vivo. (**A**) Left: image of τ (orange) and ε (blue) foci within a single *E. coli* cell, averaged over 40 s. Co-localization of the two signals is shown as a white spot. Middle: bright field image of the same cell. Right and below: fluorescence intensity of τ (orange) and ε (blue) over time. The trajectories are averaged using a 2 s moving average filter. (**B**) Averaged, normalized cross-correlation functions. The cross-correlation function of 1210 pairs of foci in living cells shows a clear positive peak (black line). The cross-correlation function for 297 pairs of foci in fixed cells (gray line) and the cross-correlation function of 1210 pairs of foci, randomized within the same field of view (red line) show no positive cross correlation. Cross-correlation functions have been vertically offset for clarity. (**C**) Exponential fit (red) to the cross-correlation function in (**B**). We obtained an exchange time scale of τ = 4 ± 2 s. The error represents the error of the fit.

The following figure supplements are available for figure 6:

**Figure supplement 1.** Growth curves for *E. coli* strains: wild-type *E. coli* (black), cells expressing both C-terminal derivatives of τ (*dnaX-YPet*) and ε (*dnaQ-mKate2*) subunits under control from their endogenous promoters (green), and cells expressing only *dnaX-YPet* (blue) and *dnaQ-mKate2* (orange).

**Figure supplement 2.** Cross-correlation analysis of simulated intensity trajectories for pairs of ε and τ foci.

**Figure supplement 3.** Measurement of concentrations of τ and ε in live cells.

counterintuitive but can be rationalized through a complex protein–protein and protein–DNA interaction network controlled and maintained by multiple dynamic interactions. Under dilute conditions, transient disruption of any one of these interactions would be followed by its rapid re-formation, preventing dissociation. If, however, there are competing Pol III*s in close proximity to the fork, one of these can bind at a transiently vacated binding site (e.g. on the $\beta_2$ sliding clamp or DnaB helicase) and consequently be at a sufficiently high local concentration to compete out the original Pol III* for binding to the other sites. Such concentration-dependent exchange has recently been reported for other systems (*Loparo et al., 2011*; *Graham et al., 2011*; *Geertsema et al., 2014*; *Gibb et al., 2014*; *Chen et al., 2015*) and mathematically described by multisite competitive exchange mechanisms (*Sing et al., 2014*; *Åberg et al., 2016*). Further evidence for a multisite mechanism comes

from comparison of the number of Pol III*s in or near the replisome at different concentrations. We quantified the number of Pol III*s at the replication fork in the in vitro experiments by normalizing the fluorescence intensity of the replisomal spot to the intensity of a single Pol III* (*Figure 4—figure supplement 3*). The peaks of the distributions are at one Pol III* per replisome (*Figure 4F*), consistent with in vivo observations (*Reyes-Lamothe et al., 2010*). Nevertheless, we find that often more than one Pol III* is present in the replisome. As its concentration increases, the binding equilibria are pushed toward occupancy of all binding sites and more than one Pol III* is associated with the replisome. At lower concentrations, Pol III* still exchanges, but the average number of Pol III*s is reduced. Our observation of Pol III* exchange in living cells shows that such a multisite exchange mechanism is a physiologically relevant pathway accessible to the replisome during coupled DNA replication. Such a mechanism may have direct implications for the mechanisms used by the replisome to deal with obstacles such as DNA damage and transcription. The ability for the replisome to rapidly exchange components in the presence of competing factors in a concentration-dependent manner could allow for components to be easily replaced from solution and provide frequent but limited access to other binding partners, such as translesion synthesis polymerases (*Sutton, 2010*), without violating fundamental chemical and thermodynamic principles.

# Materials and methods

## Replication proteins

*E. coli* DNA replication proteins were produced as described previously: the $\beta_2$ sliding clamp (*Oakley et al., 2003*), SSB (*Mason et al., 2013*), the DnaB$_6$(DnaC)$_6$ helicase–loader complex (*Jergic et al., 2013*), DnaG primase (*Stamford et al., 1992*), the Pol III $\tau_3\delta\delta'\chi\psi$ clamp loader (*Tanner et al., 2008*) and Pol III $\alpha\epsilon\theta$ core (*Tanner et al., 2008*). Highly purified *E. coli* Pol I and DNA ligase A were gifts of Yao Wang (*Wang, 2015*).

## Expression plasmids for SNAP-α

### Construction of plasmid pBOB1 (encoding w.t. α)

The *tac* promoter plasmid pND517 contains the *dnaE* gene between a pair of *Bam*HI restriction sites (*Wijffels et al., 2004*). In addition, the *Bam*HI site following the gene overlaps with an *Nco*I site such that previous digestion with *Nco*I eliminates it. To incorporate an *Nde*I site at the start codon of *dnaE*, pND517 was used as template for PCR amplification of the 5′-portion of the *dnaE* gene using primers 671 (5′-AAAA*GGATCC*TAAGGAGGTTTG*CATATG*TCTGAACCACGTTTC; *Bam*HI and *Nde*I sites are italicized, ribosome-binding site underlined) and 673 (5′-CGTTTGGCGATCTCAACGGTGT-3′). The PCR product (Fragment I; 522 bp) was isolated from an agarose gel following digestion with *Bam*HI and *Xho*I. Next, pND517 was digested with *Nco*I, and the purified linearized product digested independently with *Xho*I to generate Fragment II (3063 bp) and with *Bam*HI to yield Fragment III (5129 bp). Fragments I–III were ligated to yield pBOB1.

### Construction of plasmid pJSL2197 (encoding SNAP-α)

A modified *snap26b* gene was amplified from pSNAP-tag(T7)−2 (New England Biolabs, Ipswich, MA) by strand overlap PCR. In the first PCR, an *Nde*I site was incorporated at the start codon and an internal *Mlu*I site was removed by silent mutation using primers 728 (5′-AAAAAAAA*CATATG*GACAAAGATTGCGAA) and 729 (5′-TGAAAATAGGCGTTCAGCGCGGTCGCC), yielding Fragment I. A second PCR used primers 730 (5′-TGGCTGAACGCCTATTTTCATCAGCCGGAAGC) and 732 5′-AAAA*GGATCC*GATAGAGCCAGACTC*ACGCGT*TCCCAGACCCGG-3′) to generate Fragment II, removing the TGA stop codon and incorporating a sequence encoding a flexible peptide linker (sequence: TRESGSIGS [*Williams et al., 2002*]) flanked by *Mlu*I and *Bam*HI sites at the 3′ end of *snap26bΔMlu*I. Equimolar amounts of isolated Fragments I and II were then used as templates for PCR with the outside primers 728 and 732 to generate a product that was digested with *Nde*I and *Bam*HI and isolated from a gel. This fragment was ligated with the 502 bp *Bam*HI–*Xho*I fragment of pKO1479wt (*Ozawa et al., 2013*) encoding the N-terminal segment of α and the large *Nde*I–*Xho*I fragment of pBOB1, encoding the remainder of α, to generate pJSL2197, which directs overproduction of SNAP-α. Plasmid constructions were confirmed by nucleotide sequence determination.

## Expression and purification of SNAP-α

An affinity resin for purification of full-length (unproteolysed) α was prepared by conjugation of biotinylated $\tau_C16$ (α-binding domain V of $\tau$) (*Jergic et al., 2007*) to high-capacity streptavidin-agarose (Pierce Biotechnology, Waltham, MA). Biotinylated $\tau_C16$ (15 ml; 12 mg) was added dropwise with gentle stirring into a suspension of 6 ml of resin in 11 ml of 50 mM Tris–HCl pH 7.6, 2 mM dithiothreitol, 1 mM EDTA, 50 mM NaCl at 6°C over 20 min. Unconjugated streptavidin-agarose resin (2 ml) was added to a column and allowed to settle, then the suspension of $\tau_C16$-conjugated resin was poured over it. The column (1 × 10 cm) was then washed with 150 ml of 50 mM Tris-HCl pH 7.6, 2 mM dithiothreitol, 1 mM EDTA, 50 mM NaCl and stored at 4°C in 50 mM Tris-HCl, 5 mM dithiothreitol, 1 mM EDTA, 50 mM NaCl, 0.03% NaN$_3$.

*E. coli* strain BL21(λDE3)*recA*/pJSL2197 was grown at 30°C in LB medium supplemented with thymine (25 mg/l) and ampicillin (100 mg/l). Upon growth to $A_{600}$ = 0.6, 1 mM isopropyl-$\beta$-D-thiogalactoside (IPTG) was added and cultures were shaken for a further 3.5 hr, then chilled in ice. Cells (30 g from 6 l of culture) were harvested by centrifugation, frozen in liquid N$_2$ and stored at –80°C. After thawing, cells were lysed and SNAP-α was purified through Fraction IV essentially as described for wild-type α (*Wijffels et al., 2004*). Fraction IV (50 ml) was dialysed against two changes of 2 l of buffer Cα (25 mM Tris-HCl pH 7.6, 2 mM dithiothreitol, 1 mM EDTA, 10% (v/v) glycerol) and applied at 1 ml/min onto a column (2.5 × 12 cm) of heparin-Sepharose (*Wijffels et al., 2004*) that had been equilibrated with buffer Cα. The column was washed with 30 ml of buffer Cα and proteins were eluted using a linear gradient (150 ml) of 0–400 mM NaCl in buffer Cα. SNAP-α eluted as a single peak at ~40 mM NaCl. Fractions were collected and pooled to yield Fraction V, which was applied directly at 1 ml/min onto the column of $\tau_C16$-agarose affinity resin that had been equilibrated in buffer Dα (50 mM Tris-HCl pH 7.6, 10 mM dithiothreitol, 1 mM EDTA, 5% (v/v) glycerol) containing 20 mM MgCl$_2$. After the column had been washed with 15 ml of buffer Dα+0.6 M MgCl$_2$ and unbound proteins had been washed away, SNAP-α was eluted using a linear gradient (20 ml) of 0.6–4.0 M MgCl$_2$ in buffer Dα. SNAP-α eluted as a single peak at ~2.8 M MgCl$_2$ (*Figure 2—figure supplement 2B*). Fractions under the peak were immediately pooled and dialysed against two changes of 2 l of buffer Eα (50 mM Tris-HCl pH 7.6, 1 mM EDTA, 3 mM dithiothreitol, 100 mM NaCl, 20% (v/v) glycerol) to give Fraction VI (40 ml, containing 68 mg of protein; *Figure 2—figure supplement 2A*). Aliquots were frozen in liquid N$_2$ and stored at –80°C.

## Fluorescent labeling of SNAP-α

Two different fluorescent probes, SNAP-Surface 649 (red) and SNAP-Surface Alexa Fluor 488 (green; New England Biolabs), were used to label SNAP-α. All labeling reactions were carried out using a twofold molar excess of dye with 27 µM SNAP-α in 1 ml of 50 mM Tris-HCl pH 7.6, 2 mM dithiothreitol, 100 mM NaCl, 5% (v/v) glycerol (buffer Fα) for 2 hr at 23°C, followed by 6°C overnight with gentle rotation. Following the coupling, the reaction mixture was supplemented with 1 mM EDTA and excess dye was removed by gel filtration at 1 ml/min through a column (1.5 × 10 cm) of Sephadex G-25 (GE Healthcare, Chicago, IL) in buffer Fα+1 mM EDTA. Fractions containing the labeled SNAP-α were pooled and dialysed against 2 l of buffer Eα, frozen in liquid N$_2$ and stored in aliquots at –80°C. The degree of labeling was measured to be 90% for SNAP-α649 and 83% for SNAP-α488 by UV/vis spectrophotometry.

## Ensemble strand-displacement DNA replication assays

The flap-primed ssDNA template was made as previously described (*Jergic et al., 2013*). Conditions for the standard coupled strand extension and Pol III strand-displacement (SD) reaction were adapted from described methods (*Jergic et al., 2013*). Briefly, reactions contained 2.5 nM primed DNA template, 1 mM ATP, 0.5 mM of each dNTP, 30 nM $\tau_3\delta\delta'\chi\psi$, 150 nM Pol III (wild-type or SNAP-labeled), 200 nM $\beta_2$ and 800 nM SSB$_4$ in 25 mM Tris-HCl pH 7.6, 10 mM MgCl$_2$, 10 mM dithiothreitol and 130 mM NaCl, in a final volume of 13 µl. Components (except DNA) were mixed and treated at room temperature, then cooled in ice for 5 min before addition of DNA. Reactions were initiated at 30°C, and quenched at time points by addition of EDTA to ~100 mM and SDS to ~1%. Products were separated by agarose gel electrophoresis and stained with SYBR-Gold (Invitrogen, Waltham, MA).

## Ensemble leading and lagging strand DNA replication assays

Coupled leading and lagging strand DNA synthesis reactions were set up in replication buffer (25 mM Tris-HCl pH 7.9, 50 mM potassium glutamate, 10 mM Mg(OAc)$_2$, 40 µg/ml BSA, 0.1 mM EDTA and 5 mM dithiothreitol) and contained 1.0–1.5 nM of a 5'-biotinylated flap-primed 2 kb circular dsDNA template (Geng et al., 2011), 1 mM ATP, 250 µM CTP, GTP, and UTP, and 50 µM dCTP, dGTP, dATP, and dTTP, 6.7 nM wild-type or SNAP-labeled Pol III*, 30 nM $\beta_2$, 300 nM DnaG, 100 nM SSB$_4$, and 30 nM DnaB$_6$(DnaC)$_6$ in a final volume of 12 µl. Components (except DNA) were mixed and treated at room temperature, then cooled in ice for 5 min before addition of DNA. Reactions were initiated at 30°C, and quenched after 30 min by addition of 7 µl 0.5 M EDTA and 6 µl DNA loading dye (6 mM EDTA, 300 mM NaOH, 0.25% (v/v) bromocresol green, 0.25% (v/v) xylene cyanol FF, 30% (v/v) glycerol). The quenched mixtures were loaded into a 0.6% (w/v) agarose gel in alkaline running buffer (50 mM NaOH, 1 mM EDTA). Products were separated by agarose gel electrophoresis at 14 V for 14 hr. The gel was then neutralized in 1 M Tris-HCl, pH 7.6, 1.5 M NaCl and stained with SYBR Gold. The Okazaki fragment length distribution was calculated by normalizing the intensity as a function of DNA length.

Reactions for testing the stability of the α–τ interaction in Pol III* during replication were performed as above with modifications. First, 125 µg streptavidin-coupled magnetic beads (Invitrogen) were washed and equilibrated in replication buffer containing 200 µM AMP-PNP (replication buffer B). DnaB$_6$(DnaC)$_6$ was first loaded at the fork by incubation of 7.5 nM rolling-circle DNA and 75 nM DnaB$_6$(DnaC)$_6$ (reaction A) at 37°C for 5 min in replication buffer B (40 µl), before being immobilized on streptavidin-coupled magnetic beads for 30 min at room temperature (reaction B). Unbound DNA was removed by washing reaction B three times in 200 µl replication buffer B. Replication was initiated by resuspending reaction B in replication buffer containing 1.25 mM ATP, 250 µM CTP, GTP and UTP, 200 µM dCTP, dGTP, dATP and dTTP, 3.35 nM each of red and green labeled Pol III*, 200 nM $\beta_2$, 300 nM DnaG, 50 nM SSB$_4$, and 30 nM DnaB$_6$(DnaC)$_6$, and allowed to proceed for 20 min at 37°C. Reactions were quenched with 2.1 µl of 2.5 M NaCl and 5 µl of 0.5 M EDTA. Following quenching, the supernatant was removed, diluted 100-fold in replication buffer then imaged on the surface of a coverslip at the single-molecule level. The remaining DNA products coupled to the beads were washed three times in replication buffer, then resuspended in replication buffer (23 µl) and 7 µl DNA loading dye then heated to 70°C for 5 min. The DNA was loaded onto the alkaline agarose gel, which was run under the same conditions as before.

## In vitro single-molecule rolling-circle DNA replication assay

To construct the rolling circle template (Tanner et al., 2009), the 66-mer 5'-biotin-T$_{36}$AATTCGTAA TCATGGTCATAGCTGTTTCCT-3' (Integrated DNA Technologies, Coralville, IA) was annealed to M13mp18 ssDNA (New England Biolabs) in TBS buffer (40 mM Tris-HCl pH 7.5, 10 mM MgCl$_2$, 50 mM NaCl) at 65°C. The primed M13 was then extended by adding 64 nM T7 gp5 polymerase (New England Biolabs) in 40 mM Tris-HCl pH 7.6, 50 mM potassium glutamate, 10 mM MgCl$_2$, 100 µg/ml BSA, 5 mM dithiothreitol and 600 µM dCTP, dGTP, dATP and dTTP at 37°C for 60 min. The reaction was quenched with 100 mM EDTA and the DNA was purified using a PCR purification kit (Qiagen, the Netherlands). Microfluidic flow cells were prepared as described (Geertsema et al., 2015). Briefly, a PDMS flow chamber was placed on top of a PEG-biotin-functionalized microscope coverslip (Figure 1—figure supplement 1 inset). To help prevent non-specific interactions of proteins and DNA with the surface, the chamber was blocked with buffer containing 20 mM Tris-HCl pH 7.5, 2 mM EDTA, 50 mM NaCl, 0.2 mg/ml BSA, and 0.005% Tween-20. The chamber was placed on an inverted microscope (Nikon Eclipse Ti-E) with a CFI Apo TIRF 100x oil-immersion TIRF objective (NA 1.49, Nikon, Japan) and connected to a syringe pump (Adelab Scientific, Australia) for flow of buffer.

Conditions for coupled DNA replication under continuous presence of all proteins were adapted from previously described methods (Tanner et al., 2008, 2009). All in vitro single-molecule experiments were performed at least four times. Briefly, 30 nM DnaB$_6$(DnaC)$_6$ was incubated with 1.5 nM biotinylated ds M13 template in replication buffer (25 mM Tris-HCl pH 7.9, 50 mM potassium glutamate, 10 mM Mg(OAc)$_2$, 40 µg/ml BSA, 0.1 mM EDTA and 5 mM dithiothreitol) with 1 mM ATP at 37°C for 30 s. This mixture was loaded into the flow cell at 100 µl/min for 40 s and then at 10 µl/min. An imaging buffer was made with 1 mM UV-aged Trolox, 0.8% (w/v) glucose, 0.12 mg/ml glucose oxidase, and 0.012 mg/ml catalase (to increase the lifetime of the fluorophores and reduce blinking),

1 mM ATP, 250 µM CTP, GTP and UTP, and 50 µM dCTP, dGTP, dATP and dTTP in replication buffer. Pol III* was assembled in situ by incubating $\tau_3\delta\delta'\chi\psi$ (410 nM) and SNAP-labeled Pol III cores (1.2 µM) in imaging buffer at 37°C for 90 s. Replication was initiated by flowing in the imaging buffer containing 6.7 nM Pol III* (unless specified otherwise), 30 nM $\beta_2$, 300 nM DnaG, 250 nM SSB$_4$, and 30 nM DnaB$_6$(DnaC)$_6$ at 10 µl/min. Reactions were carried out 31°C, maintained by an electrically heated chamber (Okolab, Burlingame, CA).

Double-stranded DNA was visualized in real time by staining it with 150 nM SYTOX orange (Invitrogen) excited by a 568 nm laser (Coherent, Santa Clara, CA; Sapphire 568–200 CW) at 150 µW/cm. The red and green Pol III* were excited at 700 mW/cm$^2$ with 647 nm (Coherent, Obis 647–100 CW) and 488 nm (Coherent, Sapphire 488–200 CW) lasers, respectively (*Figure 1—figure supplement 1*). The signals were separated via dichroic mirrors and appropriate filter sets (Chroma, Bellows Falls, VT). Imaging was done with either an EMCCD (Photometics, Tucson, AZ; Evolve 512 Delta) or a sCMOS camera (Andor, UK; Zyla 4.2). The analysis was done with ImageJ using in-house built plugins. The rate of replication of a single molecule was obtained from its trajectory and calculated for each segment that has constant slope.

Conditions for the pre-assembly replication reactions were adapted from published methods (*Tanner et al., 2011*; *Georgescu et al., 2011*). Solution 1 was prepared as 30 nM DnaB$_6$(DnaC)$_6$, 1.5 nM biotinylated ds M13 substrate and 1 mM ATP in replication buffer. This was incubated at 37°C for 3 min. Solution 2 contained 60 µM dCTP and dGTP, 6.7 nM red Pol III*, and 74 nM $\beta_2$ in replication buffer (without dATP and dTTP). Solution 2 was added to an equal volume of solution 1 and incubated for 6 min at 37°C. This was then loaded onto the flow cell at 100 µl/min for 1 min and then 10 µl/min for 10 min. The flow cell was washed with replication buffer containing 60 µM dCTP and dGTP. Replication was finally initiated by flowing in the imaging buffer containing 50 nM $\beta_2$, 300 nM DnaG and 250 nM SSB$_4$ at 10 µl/min.

## Measurement of the stoichiometry of Pol III*s at the replisome

The average intensity of a single labeled Pol III core (6 pM) was calculated by immobilization on the surface of a cleaned microscope coverslip in imaging buffer. The imaging was under the same conditions as used during the single-molecule rolling-circle experiments. Using ImageJ with in-house built plugins, we calculated the integrated intensity for every Pol III core in a field of view after applying a local background subtraction. The histograms obtained were fit with a Gaussian distribution function using MATLAB 2014b, to give a mean intensity of 5100 ± 2000 for the red and 1600 ± 700 for the green Pol III core (*Figure 4—figure supplement 4*). To measure the intensity of the fluorescent spot at the replication fork, we tracked its position and integrated the intensity for both colors simultaneously over time. Given there is no decay in fluorescence intensity of labeled Pol III cores as a function of DNA length under near-TIRF imaging conditions during DNA replication (*Figure 4—figure supplement 5*), we calculated the total number of Pol III*s at every time point during coupled DNA replication by dividing these intensities by the intensity of a single Pol III*. Subsequent histograms were fit to four (6.7 nM) or three (0.3 nM) Gaussians centered at integral numbers of Pol III* (*Figure 4—figure supplement 3*) using MATLAB 2014b.

## Fluorescent chromosomal fusions

The strain EAW192 (*dnaQ-mKate2*) was constructed using a modified version of the λ RED recombination system (*Huang et al., 1997*), introducing a mutant FRT–Kan$^R$–wtFRT cassette. To select for recombinants, cells were plated on LB-agar supplemented with 40 µg/ml of kanamycin and grown overnight. Kanamycin-resistant strains were further screened for ampicillin sensitivity, to ensure that cells had been cured of the λ RED plasmid pKD46. The two-color strain EAW203 (*dnaX-YPet*, *dnaQ-mKate2*) was constructed by P1 transduction. JJC5945 cells (*dnaX-YPet*) (*Robinson et al., 2015*) were first treated with pLH29 (*Huang et al., 1997*) to remove existing Kan$^R$ markers, then infected with P1 grown on EAW192 (*dnaQ-mKate2*) cells. Transductants were selected for kanamycin resistance.

## Growth rates of fluorescent chromosomal fusions

Single colonies of wild-type *E. coli* MG1655 and derivatives containing the C-terminal chromosomal *dnaQ* and *dnaX* fusions were used to inoculate 5 ml of LB broth (with 34 µg/ml kanamycin, if

required) and grown at 37°C with shaking overnight. LB broth (100 ml) was inoculated with $1.0 \times 10^5$ cells/ml from overnight cultures. Subsequent growth of each strain was monitored at 37°C with shaking by determining $OD_{600}$ every 30 min for 9.5 hr. The doubly labeled *dnaX-YPet dnaQ-mKate2* cells grew only slightly more slowly than wild-type cells (division time = 33 ± 4 min cf. 33 ± 8 min) (*Figure 6—figure supplement 1*), indicating that labeling the ε and τ components of the replisome does not significantly disrupt DNA replication.

## In vivo single-molecule visualization assays

The cells were grown at 37°C in EZ rich defined medium (Teknova) that included 0.2% (*w/v*) glucose. For imaging, a PDMS well was placed on top of a coverslip that was functionalized with (3-aminopropyl)triethoxysilane (BioScientific, Australia) (*Robinson et al., 2015*). The cells were immobilized on the surface of the well, which was then placed on the heated stage (Okolab) of the microscope. Imaging was done at 37°C. The τ-YPet and ε-mKate2 were excited at 0.03 mW/cm$^2$ with 514 nm (Coherent, Sapphire 514–150 CW) and 3 W/cm$^2$ with 568 nm (Coherent, Sapphire 568–200 CW) lasers, respectively. The signals were separated via a beam splitter (Photometrics, DVΛ Multichannel Imaging System) and appropriate filter sets (Chroma). Imaging was done with an EMCCD camera (Photometrics, Evolve Delta). The visualization of foci in living cells was performed in triplicate. The image processing was done with ImageJ using in-house built plugins.

The concentrations or ε and τ were determined as described (*Robinson et al., 2015*) by measuring the total fluorescence within each cell and dividing by the mean intensity of a single molecule. To measure the total fluorescence, we first imaged 141 wild-type MG1655 cells to determine the cellular autofluorescence (*Figure 6—figure supplement 3A*). We found the autofluorescence to be constant for the duration of our measurements. We then imaged 273 τ-YPet, ε-mKate2 cells in 20 fields of view. The background fluorescence from the microscope coverslip was determined by fitting the photobleaching per field of view with a single-exponential decay (*Figure 6—figure supplement 3B*). The mean cellular intensities were corrected for the background and cellular autofluorescence. We then fitted individual cell photobleaching curves to obtain the amplitudes (*Figure 6—figure supplement 3C*).

To find the intensity of a single-molecule, photobleaching trajectories of single foci were determined. The τ and ε foci were identified by making average projections of movies. The intensity over time trajectories for each focus as it photobleached was measured. Next, the local background fluorescence around each focus was subtracted. Trajectories showed step-wise intensity transitions corresponding to photobleaching of single fluorescent molecules (*Figure 6—figure supplement 3D* inset). These transitions were fit by change-point analysis (*Watkins and Yang, 2005*; *Duderstadt et al., 2016*). A histogram of the step sizes showed a relatively narrow distribution (*Figure 6—figure supplement 3D*). We found the mean intensity of a single molecule by fitting with a Gaussian distribution. These were 158 ± 2 for ε and 130 ± 5 for τ (mean ± s.e.).

To find the total fluorescence intensity per cell, the mean cell intensity was multiplied by the area of the cell. This was then divided by the single-molecule intensity. It was determined that there are 104 ± 3 copies of ε and 96 ± 6 copies of τ per cell. This corresponds to a concentration of 72 ± 3 nM for ε and 67 ± 5 nM for τ. If we assume that all ε and τ are part of a Pol III* complex this tells us that there is 23 nM Pol III* in the cell. The intensities of the foci were measured by integrating the intensity of the peak and subtracting the mean local background intensity. The average cross-correlation functions were calculated using MATLAB 2014b (Mathworks). The cross-correlation was fit with an exponential decay and gave a characteristic time scale of 4 ± 2 s (mean ± error of the fit) (*Figure 6*).

## Acknowledgements

We thank Harshad Ghodke for helpful discussions and Yao Wang and Zhi-Qiang Xu for contributing reagents. This work was supported by the Australian Research Council: DP150100956 to AMvO and NED, an Australian Laureate Fellowship to AMvO (FL140100027), and Australian Government Research Training Program Scholarship to JSL.

# Additional information

## Competing interests

AMvO: Reviewing editor, *eLife*. The other authors declare that no competing interests exist.

## Funding

| Funder | Grant reference number | Author |
|---|---|---|
| Australian Government Research Training Program Scholarship | | Jacob S Lewis |
| Stichting voor Fundamenteel Onderzoek der Materie | 12CMCE03 | Lisanne M Spenkelink |
| National Institutes of Health | GM32335 | Michael M Cox |
| Australian Research Council | DP150100956 | Nicholas E Dixon<br>Antoine M van Oijen |
| Australian Research Council | FL140100027 | Antoine M van Oijen |

The funders had no role in study design, data collection and interpretation, or the decision to submit the work for publication.

## Author contributions

JSL, LMS, Conceptualization, Resources, Data curation, Formal analysis, Investigation, Visualization, Methodology, Writing—original draft, Writing—review and editing; SJ, MMC, Resources, Writing—review and editing; EAW, NPH, Resources, Conception and design of critical reagent protocols, and contribution of essential reagents; EM, Resources, Methodology; KED, Conceptualization, Software, Visualization; AR, Data curation, Software, Formal analysis, Visualization, Methodology, Writing—review and editing; NED, AMvO, Conceptualization, Resources, Supervision, Funding acquisition, Methodology, Writing—original draft, Project administration, Writing—review and editing

## Author ORCIDs

Jacob S Lewis, http://orcid.org/0000-0002-9945-6133
Lisanne M Spenkelink, http://orcid.org/0000-0002-5511-8757
Nicholas E Dixon, http://orcid.org/0000-0002-5958-6945
Antoine M van Oijen, http://orcid.org/0000-0002-1794-5161

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
