## [Decision Letter]

Thank you for submitting your article "Single-molecule visualization of fast polymerase turnover in the bacterial replisome" for consideration by *eLife*. Your article has been reviewed by three peer reviewers, one of whom is a member of our Board of Reviewing Editors, and the evaluation has been overseen by John Kuriyan as the Senior Editor. The following individual involved in the review of your submission has agreed to reveal their identity: Houra Merrikh (Reviewer #3).

Summary:

This manuscript is a well performed analysis on the exchange of components at a replication fork. An emerging picture in the field is that components of tightly associated complexes can exchange rapidly, provided an excess of components is present. This paper applies this concept to the exchange of the Pol III* at a replication fork. The exchange is studied by a single molecule approach, giving very detailed data regarding this question. Furthermore, the authors take the work into the whole cell, and find that the exchange still occurs even in the context of the cell. The data are presented clearly, and the manuscript is well written. The findings presented in this paper are fundamentally important to the field's understanding of how the replisome functions. Classically, the replisome is thought to be a stable (non-dynamic) structure, but recent work has begun to indicate that this picture is not correct. In this paper, the authors present strong evidence with regards to the dynamic nature of the replisome, potentially changing the view of how this fundamental process functions in vivo. In particular, Pol III* holoenzyme seems to rapidly exchange in and out of the replisome.

Although concentration-dependent exchange of polymerase has been reported before in the case of T7 replication machinery (this needs to be explicitly mentioned), this appears to be the first example of showing the same for *E. coli* in vitro, and we are not aware of an in vivo report, except by a recent *eLife* paper from University of Washington where they reported that the replicative helicase is disassembling and reassembling due to replication-transcription conflict. Therefore, the work is generally suitable for *eLife*. However, there are some missed opportunities, especially in solidifying the claim of concentration-dependent exchange in vivo, and more detailed summary of prior work is needed.

Essential revisions:

1) It is not clear whether the concentration dependence of the turnover rate (a key point) observed in vitro is also true in vivo. Since this concentration dependence is the main new finding, could the authors show this in live cells to strengthen this point? This is a difficult task but potential growth rate difference may modulate the concentrations of Pol III* in solution and perhaps facilitate such an analysis. Alternatively, the authors could try to conditionally deplete one of the core components and observe the dynamics as the concentrations of the proteins decrease over time. We prefer if the in vivo concentration be established experimentally. If the authors don't feel that this is necessary, then they should explain why this is the case.

2) Since the authors do not show concentration dependence in vivo, how can the authors distinguish turnover of the Pol III* complex in vivo from disassembly of the entire replisome due to potential obstacles? If the authors calculate the average rate of turnover from the intensity traces in vivo, they could potentially correlate this with the in vitro numbers to support their point.

3) They show evidence of exchange in vitro but there is no attempt to quantify the kinetics as a function of protein concentration in solution and put it in a firm mathematical footing. This is a glaring omission especially because the same group has put forward the mathematical framework.

4) It remains possible that only the cores exchange between Pol III*s at a replication fork. The control was performed in Figure 3, but in that experiment the Pol III*'s are not at a working fork with DnaB where things might be different. For example, two Pol III*'s might bind one fork (required intermediate for wholesale exchange), and this would increase their local concentration of cores to a very high level, at which time the cores might begin exchanging between the two Pol III*'s. It is not clear that the clamp loader is also getting recycled in the in vitro experiment because it is not labeled fluorescently. The lack of exchange in the context of Pol III* alone is not sufficient because one cannot rule out the possibility of the interaction becoming more dynamic in the context of replisome. It seems the better side of caution for the authors to add a statement that explains that things could possibly be different at a moving fork, such that the cores exchange among two Pol III*s. One could note, that even if this occurs, it still runs along the same message line of the paper, that seemingly tight complexes in fact are quite fluid when an excess of a component is present.

---

## [Author Response]

*Essential revisions:*

*1) It is not clear whether the concentration dependence of the turnover rate (a key point) observed in vitro is also true in vivo. Since this concentration dependence is the main new finding, could the authors show this in live cells to strengthen this point? This is a difficult task but potential growth rate difference may modulate the concentrations of Pol III* in solution and perhaps facilitate such an analysis. Alternatively, the authors could try to conditionally deplete one of the core components and observe the dynamics as the concentrations of the proteins decrease over time. We prefer if the in vivo concentration be established experimentally. If the authors don't feel that this is necessary, then they should explain why this is the case.*

We agree that changing the concentration of Pol III* is a very challenging task, especially considering that we would need to simultaneously change the concentration of all individual subunits in a controllable manner, without perturbation of other metabolic processes. We agreed that changing the growth rate by using different growth media might be a way to modulate replisomal protein concentrations. Therefore, we propagated cells containing *dnaX-YPet* and *dnaQ-mKate2* chromosomal fusions in M9 minimal medium with either glucose or acetate as the sole carbon source. It is well documented that such media decrease the growth rate of *E. coli*. For example, M9-acetate medium decreases the growth rate by ~4-fold (Pierucci (1972) J Bacteriol 109:848–854, Volkmer & Heinemann (2011) PLosONE 6: e23126). We compared the fluorescence intensities of ε and τ in these cells under the same experimental conditions used for previous in vivo measurements in EZ rich glucose medium. However, we found no difference in the intensities of either ε or τ subunits, with intensities (arb. units) of 2,053 versus 2,069 for ε and 674 versus 744 for τ in M9-glucose and acetate media, respectively. We believe that to reliably modulate the concentration of Pol III* components in vivo would require changing the entire equilibrium of protein expression in the cell. It might be possible to deplete each component of the Pol III* complex by multiple-gene silencing using antisense RNAs. However, such an approach would require a considerable investment in resources and time and we feel is outside the scope of our current study.

As requested by the reviewers, we have determined the concentrations of ε and τ in vivo under the same experimental conditions we used to establish Pol III* exchange in vivo by cross-correlation analysis. These new experiments and conclusions are described in subsection “Exchange of Pol III* complexes in live cells” and shown in Figure 6—figure supplement 3. We found that the in vivo concentrations of ε and τ are 72 ± 3 nM and 67 ± 5 nM, respectively. Assuming that all ε and τ are complexed to form Pol III*, the concentration of Pol III* would be 23 nM. This concentration is ~3-fold higher than concentrations of Pol III* in two-color in vitro experiments where rapid exchange is observed (Figure 4).

*2) Since the authors do not show concentration dependence in vivo, how can the authors distinguish turnover of the Pol III* complex in vivo from disassembly of the entire replisome due to potential obstacles? If the authors calculate the average rate of turnover from the intensity traces in vivo, they could potentially correlate this with the in vitro numbers to support their point.*

Indeed, given the complex, crowded intracellular environment, the exchange behavior within the cell could be quite different. It is difficult to predict what effect potential obstacles arising from DNA repair and recombination will have on the exchange of Pol III*. We therefore calculated the exchange time from the cross-correlation function, as the reviewers suggested. From this we found a time of 4 ± 2 s. This is of the same order of magnitude as the in vitro exchange times we found at similar Pol III* concentrations (~few seconds). We conclude that the Pol III* dynamics we observe are mostly due to Pol III* exchange. This result is now described in subsection “Exchange of Pol III* complexes in live cells” and shown in Figure 6 (previously Figure 5).

*3) They show evidence of exchange in vitro but there is no attempt to quantify the kinetics as a function of protein concentration in solution and put it in a firm mathematical footing. This is a glaring omission especially because the same group has put forward the mathematical framework.*

To address this important issue, we have extended our experiments to include in vitro single-molecule FRAP to quantify the timescales of Pol III* exchange as a function of concentration. These experiments and conclusions are described in subsection “Quantification of exchange time of Pol III* in vitro” and shown in Figure 5 and Figure 5—figure supplement 1. The main observation is that we can modulate the timescales of exchange of Pol III* in a concentration-dependent manner. At high concentrations, recovery of the fluorescence signal associated with Pol III* is rapid and at lower concentrations the recovery is much slower.

We can use the exchange times derived from in vitro FRAP experiments to quantify the kinetics as a function of protein concentration using the mathematical model described in Åberg et al., 2016. Using the mathematical framework, a *K*_D_ of 7 nM is derived, representing the average of all interactions that hold the Pol III* complex at the fork. While this number is interesting and likely tells us something about the requirements for the strengths of the interactions balancing stability and exchange, we do not have a sufficiently complete picture of all the interactions within the replisome to use this value in any meaningful interpretive manner.

*4) It remains possible that only the cores exchange between Pol III*s at a replication fork. The control was performed in Figure 3, but in that experiment the Pol III*'s are not at a working fork with DnaB where things might be different. For example, two Pol III*'s might bind one fork (required intermediate for wholesale exchange), and this would increase their local concentration of cores to a very high level, at which time the cores might begin exchanging between the two Pol III*'s. It is not clear that the clamp loader is also getting recycled in the in vitro experiment because it is not labeled fluorescently. The lack of exchange in the context of Pol III* alone is not sufficient because one cannot rule out the possibility of the interaction becoming more dynamic in the context of replisome. It seems the better side of caution for the authors to add a statement that explains that things could possibly be different at a moving fork, such that the cores exchange among two Pol III*s. One could note, that even if this occurs, it still runs along the same message line of the paper, that seemingly tight complexes in fact are quite fluid when an excess of a component is present.*

To address this important issue, we designed an experiment that allows us to image individual Pol III*s that have participated in coupled DNA replication. We performed a bulk rolling-circle replication experiment using a 8-fold excess of Pol III* over DNA, with a total Pol III* concentration of 6.6 nM. At this concentration of Pol III*, the time scale of its exchange is approximately a few seconds (Figure 5). We allowed the reaction to proceed for 20 minutes. By using this strategy, we can be confident that all Pol III*s in solution have participated in DNA replication at the fork at least several times. We then imaged these Pol III*s at the single-molecule level and did not observe any co-localization between red and green Pol IIIs. These results and conclusions are described in subsection “Exchange of Pol III* complexes in vitro” and have been added to Figure 3 and Figure 3—figure supplement 2.